# A Guided Tutorial on Modelling Human Event-Related Potentials with Recurrent Neural Networks

**DOI:** 10.3390/s22239243

**Published:** 2022-11-28

**Authors:** Jamie A. O’Reilly, Jordan Wehrman, Paul F. Sowman

**Affiliations:** 1College of Biomedical Engineering, Rangsit University, Pathum Thani 12000, Thailand; 2School of Engineering, King Mongkut’s Institute of Technology Ladkrabang, Bangkok 10520, Thailand; 3Brain and Mind Centre, University of Sydney, Sydney, NSW 2006, Australia; 4School of Psychological Sciences, Macquarie University, Sydney, NSW 2109, Australia

**Keywords:** artificial neural network, computational neurophysiology, EEG signal processing, event-related potential, P3, recurrent neural network

## Abstract

In cognitive neuroscience research, computational models of event-related potentials (ERP) can provide a means of developing explanatory hypotheses for the observed waveforms. However, researchers trained in cognitive neurosciences may face technical challenges in implementing these models. This paper provides a tutorial on developing recurrent neural network (RNN) models of ERP waveforms in order to facilitate broader use of computational models in ERP research. To exemplify the RNN model usage, the P3 component evoked by target and non-target visual events, measured at channel Pz, is examined. Input representations of experimental events and corresponding ERP labels are used to optimize the RNN in a supervised learning paradigm. Linking one input representation with multiple ERP waveform labels, then optimizing the RNN to minimize mean-squared-error loss, causes the RNN output to approximate the grand-average ERP waveform. Behavior of the RNN can then be evaluated as a model of the computational principles underlying ERP generation. Aside from fitting such a model, the current tutorial will also demonstrate how to classify hidden units of the RNN by their temporal responses and characterize them using principal component analysis. Statistical hypothesis testing can also be applied to these data. This paper focuses on presenting the modelling approach and subsequent analysis of model outputs in a how-to format, using publicly available data and shared code. While relatively less emphasis is placed on specific interpretations of P3 response generation, the results initiate some interesting discussion points.

## 1. Introduction

Electrical signals from the human brain have been recorded and reported since Hans Berger’s now-famous experiments in 1924 [1]. Since then, steady states and induced changes in electrical brain signals have been recorded using electroencephalography (EEG) and reported in several thousands of articles; e.g., searching ‘EEG’ on Google Scholar results in over 2.5 million hits. A subset of this research focuses on how the brain responds to discrete stimuli, by analyzing event-related potentials (ERPs). Various ERP patterns have been described in the literature, from the mismatch negativity (MMN), a negative difference wave that is evoked from a novel or rare stimulus [2,3,4], to the contingent negative variation (CNV), a long wave associated with readiness to respond [5,6,7,8], to the P600, a positive wave associated with grammatical or syntactic errors [9,10].

This variety of ERP components and their associated tasks has led to an ever-increasing number of possible analyses, from multivariate pattern analysis (MVPA) and principal component analysis (PCA) to peak amplitude correlations and source localization methods. However, characterizing the neural sources that give rise to these ERP components and transform sensory inputs into encoded neural representations remains challenging. Recurrent neural networks (RNNs) offer a novel approach in this respect [11,12]. Neural networks are machine learning techniques that receive input data (e.g., an image of a face) and submit them to a series of hidden layers to arrive at an output (e.g., an ERP). By iteratively training a neural network, the weights connecting individual ‘neurons’ within and between layers are updated, and the output progressively resembles the training data. Once the neural networks are adequately trained, several possible analyses are available, such as feeding the neural network unseen data or analyzing the properties of hidden layers. 

In terms of time-series data generally, or EEG specifically, RNNs provide a tool for various analyses. Aside from conventional uses, such as classification of an ERP as coming from one type of task or another or predicting what input (e.g., sounds) was played to a person based on their ERPs, similarities in the neural patterns elicited by different paradigms not previously recognized could be established. For example, an RNN trained on data from one experiment could be used to predict ERPs from another paradigm, revealing possible similarities between them. This approach could be used as a hypothesis generator; by training an RNN on one set of data from a limited group of participants, predictions could be made regarding novel inputs. More deeply, by training multiple RNNs on various ERPs, hidden layer dynamics could be analyzed to reveal how the neural network acts in similar or disparate ways to make sense of the data. 

Technical challenges may deter researchers from exploring these novel possibilities, restricting the application of RNNs in EEG research. While various Python, R and MATLAB packages exist to make implementation relatively straightforward, a simple introduction to their use has yet to be provided in the context of EEG research.

In the current article, we aim to fill this gap in training. We offer a simple-to-use, step-by-step tutorial with data readily available online through the ERP CORE (Compendium of Open Resources and Experiments) website (https://erpinfo.org/erp-core; accessed on 1 October 2022), which will get the interested researcher up and running in relatively short order. The current article will show the reader, through an associated Python notebook, how to model P3 ERPs from an active visual oddball paradigm using RNNs, cross-validate the findings and analyze hidden layers from the RNN to make inferences about brain activity evoked during this behavioral task. 

The P3 is a positive wave occurring approximately 300 ms after the onset of a salient stimulus. In the active oddball task, the evoked P3 wave is larger for rare target stimuli than for frequent non-targets [13,14,15,16,17]. This component has attracted much scientific interest, from its association with various neuropsychological conditions [18,19,20] to its use in brain–computer interfaces [21,22]. Further, this component can be readily found in humans and has been well characterized [23], making it an ideal learning dataset that can be readily replicated in most laboratories. 

This tutorial will follow a logical format, starting by introducing the tools needed to follow the tutorial. Though we did not perform the preprocessing of the EEG, we will briefly go through the way the data has been handled in ERP CORE and move on to a step-by-step walkthrough of how to model the data using an RNN and analyze and validate the output. This tutorial does require some basic Python programming knowledge; however, it should not be overly difficult to follow for most researchers that have programmed before.

## 2. Materials and Methods

### 2.1. Coding Environment

The code accompanying this tutorial is available from https://github.com/Ajarn-Jamie/rnn-p3-erp-tutorial (accessed on 14 October 2022). The results presented below were obtained from a computer running Windows 11 with Python 3.10.2 and the following third-party modules: Jupyter Notebook 6.4.12 [24], Matplotlib 3.5.2 [25], MNE 1.0.3 [26], NumPy 1.22.4 [27], Scikit-learn 1.1.1 [28], SciPy 1.8.1 [29] and TensorFlow 2.9.1 [30]. These dependencies can be installed using the “pip install-r requirements.txt” command.

### 2.2. Data and Preprocessing

We are modelling data from the active visual oddball P3 experiment provided as part of the ERP CORE database [31]. This data can be fetched automatically using wget, then decompressed using tar and stored in a data directory. Alternatively, the database can be acquired by clicking on the appropriate links in the ERP CORE repository. The downloaded data includes raw and preprocessed recordings from Kappenman et al. (2021) [31]. We use the preprocessed recordings that have already been re-referenced to the average of channels P9 and P10 (located near the mastoids), high-pass filtered with a cut-off frequency of 0.1 Hz and have had gross artifacts removed. Eye-movement-related artifacts were removed using independent component analysis. This data is further low-pass filtered with a cut-off frequency of 20 Hz and resampled to 100 Hz before segmenting recordings from −0.2 to 0.8 s around each event and performing average baseline correction. Only data from channel Pz is retained for further analysis and modelling.

In the active visual oddball experiment, images of letters were shown to subjects while their EEG was recorded using a Biosemi ActiveTwo system with 30 electrodes. The letters A, B, C, D and E were included, and subjects were instructed to respond to only one of these letters (the ‘target’) per block with a button press. All other letters were responded to with a different button. Subjects performed five blocks, so each letter was the target in one block.

For each stimulus block, the five letters were shown randomly with equal probability, for 0.2 s at a time, with an interstimulus interval of 1.2 to 1.4 s. Each subject was presented with 200 trials across five stimulus blocks (i.e., 40 per block), resulting in 25 events that can be separated into two categories: 5 target events (i.e., AA, BB, CC, DD and EE), where the presented stimulus matched the target, and 20 non-target events (i.e., AB, AC, AD, etc.), where the presented stimulus did not match the target. Throughout this paper, events are occasionally referred to using this convention, where the first letter represents the target and the second letter represents the stimulus presented. Stimulus pairs consisting of matching letters are considered target events, and pairs composed of different letters are non-target events.

Six subjects were removed from the dataset because they met one or more of the following criteria: >25% rejected trials due to artifacts, <50% of target trials remaining after artifact rejection, or they had a total accuracy, in terms of correct button presses in response to target and non-target stimuli, of less than 75%. Thus, a dataset containing 34 subjects, each with 25 average ERP waveforms and an epoch duration of 1.2 s (sampled at 100 Hz) for a single channel is obtained, which can be ordered into a multidimensional array of shape (34 [subjects], 25 [waveforms], 121 [samples], 1 [channel]).

### 2.3. Modelling

#### 2.3.1. Inputs and Output Labels

Input representations are matrices of size (121 [samples], 2 [conditions]), where the first dimension is the sequence length, equivalent to the number of time samples in the ERP epoch, and the number of event types determines the second dimension. Target and non-target events are represented by setting the respective first-dimension index high (=1) during the time samples where the stimulus was shown (i.e., from 0 to 0.2 s), as illustrated on the left side of Figure 1.

For supervised learning, these input representations are paired with labels derived from channel Pz data. There are 34 subjects, each with 25 ERPs (5 target and 20 non-target events), producing 850 input representation-ERP pairs used for training. These ERP waveform labels are rescaled to normalized units so that ±1 μV = ±1. The RNN parameters are fitted to minimize mean-squared error (MSE) loss between model outputs and ERP labels. By optimizing for MSE, model outputs effectively approximate the grand-average ERP waveforms for the two input conditions.

#### 2.3.2. Model Architecture and Training

The model is constructed around the SimpleRNN layer class in TensorFlow. The architecture consists of four hidden layers, each comprising 64 units with the rectified linear unit (ReLU) activation function applied, and an output layer with a single recurrent unit and linear activation function. Forward connection weights are initialized from the Glorot uniform distribution [32], recurrent weights are initialized from an orthogonal distribution [33] and biases are initialized as zeros. The model is optimized using the back-propagation-through-time algorithm implemented in TensorFlow, with a batch size of 100, for 1000 epochs. The code accompanying this article can easily be modified to customize the model architecture and initialization parameters.

Repeatable, deterministic behavior of the model is achieved by setting the random seed for stochastic algorithms, such as the weight initializers. Multiple models may be trained with stochastic behavior for other purposes and then compared to determine the best-fitting model (e.g., as in [11]).

#### 2.3.3. Cross-Validation and Final Model

A five-fold cross-validation procedure is performed to evaluate the suitability of the RNN for modelling grand-average ERP waveforms. Five models are initialized and trained using a dataset excluding one target event and one non-target event. Specifically, the validation data removed from the whole dataset for each fold consists of target events AA, BB, CC, DD and EE paired with non-target events EB, AC, BD, CE and DA. After training the model using the hyperparameters described above, it is used to predict the response for these validation data. The results are plotted in Figure 2. For both target and non-target ERP waveforms, cross-validation model outputs are highly correlated (Pearson’s r^2^ > 0.95), matching the average correlation between individual event ERPs with the grand-average event-type ERP (r^2^ = 0.988 for target, and r^2^ = 0.976 for non-target).

After cross-validating the model architecture, a final model is trained using all available data. This final model’s output and hidden unit responses can be analyzed to explore how it reproduces waveforms matching those evoked by different event types. Doing so provides a computational tool for interpreting the process of ERP generation.

#### 2.3.4. Analysis of Model Behavior

Hidden unit activations are plotted as matrix images in Figure 3, where coloring indicates hidden unit output value. Hidden units are classified according to their peak latency in response to target and non-target input representations, as detailed in Table 1. The time bins selected for these classifications are no-peak (i.e., silent or redundant unit), <0.0 s, 0.0 to 0.1 s, 0.1 to 0.2 s, 0.2 to 0.3 s, 0.3 to 0.4 s, 0.4 to 0.6 s and >0.6 s.

Hidden unit activations are plotted as time-domain signals in Figure 4, with traces colored according to their classification. This figure presents model responses to target and non-target input conditions. Their difference is computed by subtracting responses to the non-target input condition from the target input condition.

Principal component analysis (PCA) is used to compare model responses to target and non-target input conditions and their difference. Time-domain hidden unit activations from each layer, matrices of size (64 [units], 121 [samples]), are transformed into principal component spaces of size (64 [units], 2 [components]). This transformation preserves as much variance as possible from data in the original matrix while compressing them into fewer columns. Representing 121 time samples with only 2 principal components allows this data to be visualized on a cartesian plane. The results from PCA are plotted in Figure 5, with data points colored according to hidden unit classifications by peak latency described above.

Peak amplitudes or other measurements from hidden unit activations can be evaluated using statistical methods. Two approaches for comparing peak amplitudes are taken here: (i) by latency range–layer pairs containing more than two units for both target and non-target conditions, and (ii) by latency range across all layers. Independent t-tests with Bonferroni–Holm corrections were performed to establish whether differences between hidden unit peak amplitudes in response to target and non-target inputs were statistically significant, with an alpha of 0.05.

## 3. Results

### 3.1. Cross-Validation Performance

Outputs from models trained during five-fold cross-validation are plotted in Figure 2. In each fold, a ‘new’ RNN was initialized and optimized to replicate the grand-average ERPs of its training labels. Thereafter, two validation ERPs (one target and one non-target event) left out from the training dataset were compared with the outputs generated by the trained model. This was repeated five times using a different pair of target and non-target validation ERPs, as described above. Over the five folds, model outputs and cross-validation ERPs were highly correlated (all r^2^ > 0.97). This approach is consistent with some of the core principles of ERP analysis, as events are grouped into distinct categories, and the average waveform produced by these events reflects a stereotypic brain response. Thus, training the model to reproduce the grand-average of a sample of training ERPs and then comparing this output with some left-out validation ERPs verifies (i) that the model reproduces the grand-average ERP and (ii) that events within the same category produce ERPs that are strongly correlated with the grand-average. After verification, the model can be trained using the entire dataset to obtain a more complete representation of the whole-study grand-average ERP waveforms.

### 3.2. Hidden Unit Colormaps

The performance of the final model, trained using the whole dataset, is visualized in Figure 3. The upper four rows display colormap representations of hidden unit activations in response to target and non-target input conditions. The bottom row displays model outputs plotted alongside grand-average ERP waveforms. Viewing this figure from top-down, i.e., from layer 1 to layer 4, it appears that information propagates through the layers, evoking waves of activation across multiple hidden units that spread in complexity, peak latency and amplitude. In some respects, this is analogous to sensory information processing in the ascending central pathways. Magnitude differences between responses to target and non-target events also feature prominently across all model layers. In addition, the outputs shown in the bottom row seem to demonstrate a slightly better fit to non-target waveforms, which might be due to there being a higher number of non-target events (20 [events] × 34 [subjects]) than target events (5 × 34) in the training dataset.

### 3.3. Hidden Unit Categorization

Hidden unit classification based on peak latency range is detailed in Table 1. Similar levels of model redundancy, quantified by the numbers of hidden units without any change in activity (i.e., the *no-peak* category), are observed from responses to target and non-target input conditions. Specifically, across all layers of the model, target and non-target inputs produced comparable numbers of hidden units without any change in activity (i.e., T = 62 and NT = 60). This redundancy presents an analogy to the sparseness of representations in biological neural networks of the auditory and visual cortices [34,35,36]. 

Non-target inputs produced higher numbers of units with peak latency from 0 to 0.1 s (i.e., T = 36 and NT = 64 ). This pattern reversed in the 0.2 to 0.3 s window, where target input caused more hidden units to peak than non-target input (i.e., T = 47 and NT = 17), reflecting the onset phase of the P3 component. The numbers of units peaking later than 0.3 s were more comparable for both input conditions, perhaps indicating that the sources of ERP differences across the 0.3 to 0.6 s latency range are higher amplitudes evoked by the target condition rather than a greater number of sources recruited.

Signals from layer 4 units are weighted and combined to produce model output that matches the grand-average ERP. This provides some rationale for interpreting these signals as akin to those that may be produced by neural sources that generate the ERP waveform. Viewing the layer 4 hidden unit categorizations in Table 1, it can be seen that target and non-target inputs produced similar numbers of units in each category. However, perhaps it is noteworthy that non-target input produced no units peaking across the 0.2 to 0.4 s windows, while target input produced three units peaking inside this range.

### 3.4. Dynamics of Hidden Unit Activations

Hidden unit activations are plotted in the time domain in Figure 4, which presents more detailed visualizations of their time-varying dynamics than the colormap images in Figure 3. From the earliest information-processing stages, layer 1 shows notable differences between model responses to target and non-target. This difference partially results from the manner in which input conditions were represented, using a different channel to encode target and non-target events. However, a large portion of the difference between the two input conditions occurs after stimulus offset (at 0.2 s) of the target input condition, which may be more likely to reflect fundamental differences in sequential information processing between the two conditions. This distinct response to target inputs advances through the network layers, leading to a series of large-amplitude hidden unit activations within the 0.4 to 0.6 s window that contribute towards the P3 feature reproduced by the model.

Layer 4 hidden unit peak latencies in response to target and non-target input conditions favor the 0.4 to 0.6 s range (23 units for target and 22 units for non-target) over the 0.3 to 0.4 s range (2 and 0 units, respectively). In contrast, the difference between layer 4 hidden unit activations, plotted on the bottom row of Figure 4, displays more balance between the number of units that peak in the 0.3 to 0.4 s (8 units) and 0.4 to 0.6 s (16 units) ranges. Peaks in the difference between responses to target and non-target inputs therefore span the recommended measurement window for P3 of 0.3 to 0.6 s [31], whereas responses to either input condition alone tend to peak during the 0.4 to 0.6 s measurement window. This signal pattern in hidden unit difference waves reflects faster onsets and larger amplitudes of responses to target input.

Moreover, non-target input induces greater alpha-frequency (i.e., 8 to 13 Hz) oscillatory activity in layer 4 hidden units compared with target input, which is observed from 0.2 and 1 s post-stimulus. This spontaneous behavior of the model presents parallels to observations from electrophysiology recordings and is consistent with findings which demonstrate that target stimuli presented in the visual active oddball paradigm suppress central alpha-wave activity [37,38].

### 3.5. Principal Component Analysis

Results from PCA displayed in Figure 5 highlight principal sources of variance among hidden unit responses to the two input conditions and the difference between them (i.e., *target–non-target*). The two principal components shown accounted for between 68% and 95% (mean 93.49%) of the variance in each unit-by-time matrix. Coloring datapoints by respective hidden unit peak latency categorization allows us to identify what groups of units are responsible for relative proportions of variance in different layer hidden units. For instance, in layer 1, units peaking in the 0.2 to 0.3 s and 0.3 to 0.4 s ranges feature more prominently in response to target input. 

### 3.6. Statistical Analysis

Data obtained from hidden unit activations are amenable to statistical testing. For example, peak amplitude or area under the curve of hidden unit activations in different layers can be measured and compared. However, this form of statistical analysis is limited in cases where an individual hidden unit dominantly contributes to a feature of the ERP waveform, which may equally apply to analysis of biological sources of ERP components. For the present study, peak amplitudes of units were analyzed with independent t-tests, followed by Bonferroni–Holm corrections. Comparisons between target and non-target responses at specific latency range and layer pairings (19 tests) returned statistically significant differences between layer 1 units peaking in the 0.2 to 0.3 s range [t(39) = 5.289, *p* = 9.021 × 10^−5^] and layer 2 units peaking in the 0.3 to 0.4 s range [t(26) = 5.382, *p* = 0.0002]. Furthermore, comparisons between hidden unit peak amplitudes elicited in each latency range across all layers (7 tests) returned statistically significant differences between target and non-target responses in the 0.2 to 0.3 s [t(62) = 3.991, *p* = 0.0009] and 0.3 to 0.4 s [t(52) = 4.048, *p* = 0.001] latency ranges.

## 4. Discussion

During the optimization process, referred to as “training”, weights applied to connections between computational units of the RNN are adapted to bring its outputs into close agreement with grand-average ERP waveforms. Despite widespread acknowledgement that backpropagation algorithms used to train artificial neural networks are biologically infeasible, trained networks may nevertheless provide helpful analogies for interpreting how neural representations are formed [39,40].

Studying how the RNN produces outputs matching grand-average ERP waveforms could thus provide insights into the computational principles that underpin ERP generation. As such, these models can be used for interpreting data and advancing hypotheses for further research. For example, recurrent neural networks have been used to study motor pattern learning by modelling the trajectories of three-dimensional avatars derived from videos of human movement [41]. Similarly trained models, such as convolutional neural networks and autoencoders, have been constructed to study auditory pattern learning in auditory cortex neurons [42,43], and top-down control of auditory processing by attentional modulation [44]. There are open-source software tools designed specifically for applying these types of models to decode EEG/MEG data [45]. Analytical techniques, such as representational similarity analysis [46], can be applied to compare responses of these kinds of computational models with those observed of brain signals, facilitating this area of study.

After the model is optimized, the signals that it generates may be viewed as a proposed solution to the inverse source problem, where individual hidden units are considered to resemble the activity of populations of neurons. This interpretation is partly supported by the multiple respects in which the model behaves analogously to established neurophysiological phenomena. For example, information flows through hierarchically organized layers, resembling sensory processing. These signals are transferred through a network of interconnected units, representing biological neurons, that also provide feedback to each other, comparable with biological neural networks. Activations produced by successive layers of the model increase in complexity, which may bear some resemblance to neural signals that travel from sensory periphery to the brainstem, midbrain, primary, secondary and tertiary cortical areas [47,48]. Model units also spontaneously exhibit oscillatory behavior, as seen in layer 4 units in Figure 3 and Figure 4, which is comparable with oscillations that feature prominently in electrophysiological recordings from neural tissue [49,50]. Some of these features are imposed by structural constraints (hierarchical layers, recurrent computational units), whereas others (pattern of signal complexity changes, oscillations) emerge after training the model.

In one sense, the RNN modelling approach described above is a dimensionality increasing technique, contrasted with dimensionality reduction techniques such as PCA. In machine learning applications, it is often desirable to compress or derive abstract representations of data that preserve the main sources of variance for downstream inference. However, one core tenet of ERP theory is that scalp-recorded potentials are produced by many underlying components, conceptualized as current dipoles, originating within cortical and subcortical structures [51]. In this context, it may be beneficial to extrapolate multiple signals that represent these assumed sources, thus increasing the number of dimensions from 1 (channel Pz in this example) to 256 (obtained from hidden unit activations). Signals produced by hidden units in the penultimate layer are foremost analogous to neural sources conventionally thought to generate ERP waveforms, given that they are directly mapped onto the model output unit response. These would make promising candidates for transformation into localized source coordinates.

There are important limitations to address. Firstly, the approach taken here is incapable of modelling trial-to-trial dynamics of brain responses to experimental events. Instead, the model depicts a static, grand-average response consistent with conventional ERP analysis. In order to characterize intertrial dynamics, model input representations could be modified to encode trial order prior to training, similar to the way that state changes have been modelled [52]. Secondly, RNN performance varies considerably with stochastic weight initialization and optimization algorithms. This can result in diverse solutions to reproduce ERP waveforms that require either adjudication (finding the best-fitting model) or aggregation (finding common patterns in models) evaluation methods. The present tutorial has illustrated how this can be avoided and deterministic performance enforced by setting the seeds for random number generators used in weight initialization and optimization algorithms.

## 5. Conclusions

This tutorial has covered how to model ERP waveforms using an RNN, with the P3 response evoked by an active visual oddball experiment taken as an example. Some analysis methods have been presented, although these are not exhaustive, and more techniques could be applied to data obtained from this model. Conceptually, model hidden units resemble components that combine to produce ERP waveforms. Numerous analogies between model behaviors and physiological observations suggest that this approach could be fruitful in interpreting the computational principles underlying ERP generation.

## Figures and Tables

**Figure 1 sensors-22-09243-f001:**
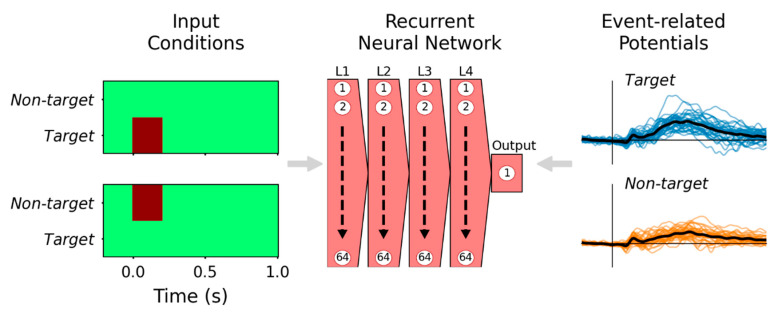
Supervised learning paradigm for training the RNN to reproduce ERP waveforms. The active oddball experiment had two fundamental event types, represented by two-element time sequences (**left**): target (**left-upper**) and non-target (**left-lower**) events; the darker color represents values of 1, but otherwise, these input representations were 0. Event-related potential waveforms evoked by each condition from subjects in the ERP experiment (**right**) were provided to the RNN as labels during training. After training, the model effectively reproduces the grand-average ERP of respective event types, plotted with black traces on the right-hand side. The recurrent neural network diagram (**center**) represents four hidden layers (L1–L4), each with 64 recurrent units, and a single recurrent output unit.

**Figure 2 sensors-22-09243-f002:**
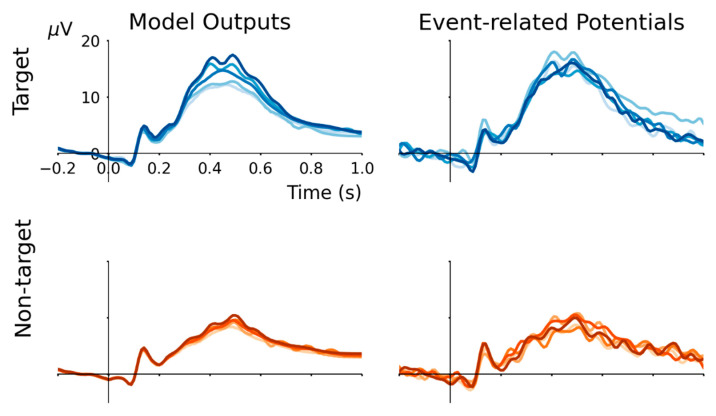
In five-fold cross-validation, model outputs strongly correlate with held-out validation ERPs. Target events AA, BB, CC, DD and EE paired with non-target events EB, AC, BD, CE and DA were held-out in turn to compute validation ERP waveforms that were compared with model outputs. Because the trained RNN produces output that approximates the grand-average ERP from each input condition, and the ERP evoked by an individual event does not deviate substantially from the grand-average of its event-type (i.e., target or non-target), the model outputs are highly correlated with held-out ERPs (all r^2^ > 0.95).

**Figure 3 sensors-22-09243-f003:**
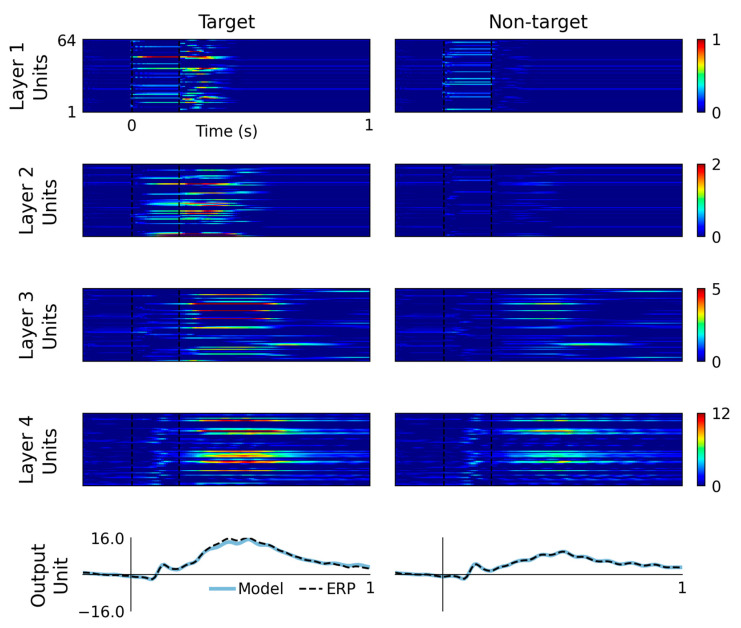
Model outputs and hidden layer activations are plotted by layer and input condition. These graphs display the pattern of information flow through the RNN, from layer 1 through to layer 4, with increasing complexity and latencies of activity, culminating in model outputs plotted alongside grand-average ERP waveforms. Target events evoke a higher magnitude response from the earliest processing layers, which leads to a higher magnitude response over the P3 latency range of 0.3 to 0.6 ms.

**Figure 4 sensors-22-09243-f004:**
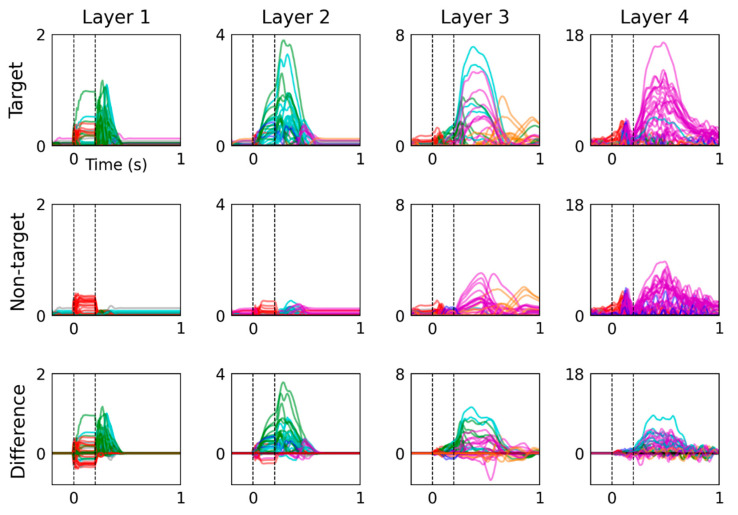
Time-domain analysis of hidden unit activations reveals distinct activity patterns in response to target and non-target input conditions. Hidden units were categorized by peak latency and their traces are colored as follows: no-peak (black), <0 s (grey), 0 to 0.1 s (red), 0.1 to 0.2 s (blue), 0.2 to 0.3 s (green), 0.3 to 0.4 s (cyan), 0.4 to 0.6 s (magenta) and >0.6 s (orange). The difference between hidden unit activations (bottom) was calculated by subtracting the responses to non-target input from those elicited by target input. Hidden unit class in the bottom row was determined by absolute peak latency.

**Figure 5 sensors-22-09243-f005:**
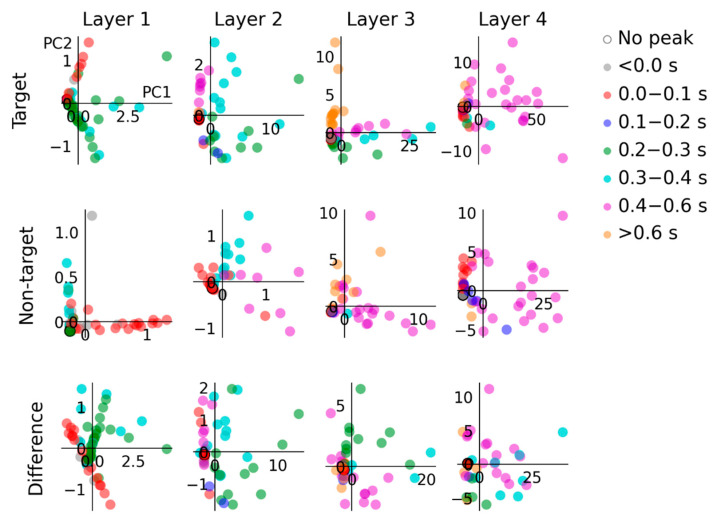
PCA highlights different sources of model variance in response to target and non-target input conditions. In layer 1, the target condition exhibits a dominant source of variance from units that peak from 0.2 to 0.3 s and 0.3 to 0.4 s, whereas the main source of variance in responses to the non-target condition are due to units that peak from 0 to 0.1 s. In layer 2, variance in the target condition is mainly due to units peaking from 0.2 to 0.3 s, 0.3 to 0.4 s and 0.4 to 0.6 s; in contrast, the non-target condition variance primarily arises due to units peaking from 0.3 to 0.4 s and 0.4 to 0.6 s. In layer 3, model responses to both conditions are similarly influenced by units peaking >0.6 s. In layer 4, the units that contribute directly to the model output show a similar pattern of principal components in response to target and non-target inputs. The main sources of variance are due to units that peak from 0.4 to 0.6 s.

**Table 1 sensors-22-09243-t001:** Model hidden unit classifications by peak latency.

Latency Range	Layer 2		Layer 3		Layer 3		Layer 4		Total	
	T	NT	T	NT	T	NT	T	NT	T	NT
No peak	14	13	20	19	11	11	17	17	62	60
<0.0 s	2	9	4	3	4	4	4	6	14	22
0.0 to 0.1 s	7	21	6	17	13	15	10	11	36	64
0.1 to 0.2 s	0	0	2	0	0	3	5	6	7	9
0.2 to 0.3 s	28	13	9	2	9	2	1	0	47	17
0.3 to 0.4 s	10	8	13	15	5	1	2	0	30	24
0.4 to 0.6 s	3	0	9	8	10	20	23	22	45	50
>0.6 s	0	0	1	0	12	8	2	2	15	10
**Total**	64		64		64		64		64	

## Data Availability

The data used in this study are openly available from https://doi.org/10.18115/D5JW4R (accessed on 1 October 2022) and the tutorial code can be accessed from https://github.com/Ajarn-Jamie/rnn-p3-erp-tutorial (accessed on 14 October 2022).

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
