# Peer review of "A Guided Tutorial on Modelling Human Event-Related Potentials with Recurrent Neural Networks"

_sensors, 2022, doi:10.3390/s22239243_

Round 1
Reviewer 1 Report
A guided tutorial on modelling human event-related potentials 2 with recurrent neural networks
This article presents a tutorial for implementing recurrent neural networks (RNN) for the temporal modeling of event-related potentials (ERP). As a result of the method, the classification of the different hidden units according to their response time is presented. the execution of the prediction algorithm to validate its interaction in the modeling process.
In general, it is an interesting work, well written, although it is recommended to double-check the grammar and spelling.
Mainly I have a few doubts that I would like the authors to address:
1. The Link provided in section 2.1 is not working anymore (https://github.com/Ajarn-Jamie/rnn-p3-erp-tutorial), then I couldn’t follow the tutorial to verify its utility and try to replicate the results
2. In section 2.2 I feel as if it is necessary to include a block diagram that allows to understand better the database acquisition protocol followed by the researchers.
3. In the methodology, they talk about the use of PCA and ICA for comparison between target and non-target inputs, and denoising signals respectively, but it is necessary to include a better description of how they are used for these purposes, please provide a bit of theoretical background of the methods and how do you use them.
4. Please provide more information or a brief description of the content in table 1, highlighting the most relevant information of the table, it is introduced in section 2.3.4 but its content is not easy to understand.
Reviewer 2 Report
General comment:
This manuscript describes a guided tutorial on modelling human event-related potentials using recurrent neural networks. The work is relevant in the field of ERP towards providing data analytics methods. Furthermore, tutorials on the field of RNNs and its applications, could strengthen its application in real-world scenarios. The conceptual framework is clear and the results are well supported. The manuscript is interesting and well written. I have some points that should be addressed before the manuscript can be accepted.
Comment 1:
It should be useful to provide a comparison metric or summary of the proposed tutorial against other available in the state-of-the-art.
Comment 2:
Regarding the architecture of the RNN, it could be nice to have a diagram of it. Hence, the readership can have a detailed overview of the RNN at a glance.
Comment 3:
It is not clear the role of PCA within the method. In the machine learning chain, PCA (unsupervised learning) must be prior RNN (supervised learning) for labeling the data. Please, clarify this situation.
Round 2
Reviewer 1 Report
Thanks to the authors for address the suggestions,